# Rapid carbon accumulation at a saltmarsh restored by managed realignment exceeded carbon emitted in direct site construction

**Hannah L. Mossman**[1]*, **Nigel Pontee**[2,3], **Katie Born**[4], **Colin Hill**[1], **Peter J. Lawrence**[1,5,6], **Stuart Rae**[1], **James Scott**[7], **Beatriz Serato**[2], **Robert B. Sparkes**[1], **Martin J. P. Sullivan**[1], **Rachel M. Dunk**[1]

**1** Ecology and Environment Research Centre, Department of Natural Sciences, Manchester Metropolitan University, Manchester, United Kingdom, **2** Jacobs, Bristol, United Kingdom, **3** University of Southampton, Southampton, United Kingdom, **4** Jacobs, Edinburgh, United Kingdom, **5** School of Ocean Sciences, Bangor University, Menai Bridge, United Kingdom, **6** Institute of Science and Environment, University of Cumbria, Ambleside, United Kingdom, **7** Jacobs, Exeter, United Kingdom

* h.mossman@mmu.ac.uk

**Data Availability Statement:** The data are available at DOI: 10.5281/zenodo.7199417.

**Funding:** PJL was funded by a Manchester Metropolitan University PhD studentship. The

## Abstract

Increasing attention is being paid to the carbon sequestration and storage services provided by coastal blue carbon ecosystems such as saltmarshes. Sites restored by managed realignment, where existing sea walls are breached to reinstate tidal inundation to the land behind, have considerable potential to accumulate carbon through deposition of sediment brought in by the tide and burial of vegetation in the site. While this potential has been recognised, it is not yet a common motivating factor for saltmarsh restoration, partly due to uncertainties about the rate of carbon accumulation and how this balances against the greenhouse gases emitted during site construction. We use a combination of field measurements over four years and remote sensing to quantify carbon accumulation at a large managed realignment site, Steart Marshes, UK. Sediment accumulated rapidly at Steart Marshes (mean of 75 mm yr$^{-1}$) and had a high carbon content (4.4% total carbon, 2.2% total organic carbon), resulting in carbon accumulation of 36.6 t ha$^{-1}$ yr$^{-1}$ total carbon (19.4 t ha$^{-1}$ yr$^{-1}$ total organic carbon). This rate of carbon accumulation is an order of magnitude higher than reported in many other restored saltmarshes, and is somewhat higher than values previously reported from another hypertidal system (Bay of Fundy, Canada). The estimated carbon emissions associated with the construction of the site were ~2–4% of the observed carbon accumulation during the study period, supporting the view that managed realignment projects in such settings may have significant carbon accumulation benefits. However, uncertainties such as the origin of carbon (allochthonous or autochthonous) and changes in gas fluxes need to be resolved to move towards a full carbon budget for saltmarsh restoration.

funders had no role in study design, data collection and analysis, decision to publish, or preparation of the manuscript.

**Competing interests:** The authors have declared that no competing interests exist.

## Introduction

Earth's ecosystems take up more carbon from the atmosphere than they release into it, causing increases in atmospheric $CO_2$ to be smaller than expected from fossil emissions and land-use change [1]. They can also contain substantial carbon stocks, largely derived from atmospheric carbon, and these stocks are sensitive to changes in climate or land-use [2, 3]. Coastal 'blue carbon' ecosystems, including saltmarshes, are especially carbon dense and sequester carbon at a faster rate per unit area than terrestrial ecosystems [4]. Carbon accumulates in salt marshes through both the deposition of sediment and organic matter carried in by the tides (allochthonous carbon) and through in-situ plant growth (autochthonous carbon). Globally, the ~5.5 million hectares of saltmarshes [5] are estimated to accumulate carbon at an average rate of ~2.4 t C ha$^{-1}$ yr$^{-1}$ [6]. Despite their large carbon stocks, ~50% of saltmarsh area has been lost, particularly through reclamation for agriculture or urbanisation, or degraded by pollution, invasive species and hydraulic alteration [7, 8], with annual losses of 1–2% [9, 10].

In response to losses of saltmarsh and its associated biodiversity, 'no net loss' policies have sought to protect remaining wetlands, restored degraded sites and create new habitat [11], contributing to over 100,000 ha of intertidal wetland restoration or creation over the last 30 years [12]. However, the pace of global wetland creation or restoration is not sufficient to offset losses, where a key barrier is the availability of project financing [13]. Payments for ecosystem services, such as flood protection or biodiversity, offer potential financial mechanisms for saltmarsh creation or restoration [14]. Carbon accumulation (and thus climate mitigation) has been recognised as a potential benefit of saltmarsh restoration, and could therefore provide a further motivation for site creation or restoration [15, 16].

Quantifying the rate of carbon accumulation in restored saltmarshes will be necessary if carbon finance mechanisms are to be developed [17] and is also important to enable saltmarsh restoration to be properly included in national carbon budgets [18]. Furthermore, rising sea levels threaten existing saltmarshes, and the climate sensitivity of their carbon stocks and fluxes needs to be quantified [19]. While saltmarsh restoration could potentially compensate for loss of natural saltmarshes, given known differences in topography and ecology [20, 21], it may not be appropriate to assume that restored or created marshes will ultimately store carbon at a rate comparable to natural saltmarshes [22]. Furthermore, the methods used in site restoration will also likely affect the total carbon sequestration and/or the rates of accumulation. For example, in mangroves, naturally and artificially regenerated forests differ in their structure, tree diversity and regeneration rate [23], potentially leading to differences in carbon storage rates, and in saltmarshes sedimentation rates differ between sites restored by managed realignment compared to regulation of tidal inundation [24]. It is therefore important to determine any differences between carbon accumulation in natural and restored saltmarshes, and any differences between restoration techniques.

Previous attempts to quantify actual or potential carbon accumulation following saltmarsh restoration have used a variety of techniques: (a) spatially explicit models to predict landscape-scale carbon accumulation based on observed carbon accumulation in natural habitats [25]; (b) measurements at a single time-point to take a snapshot of carbon stocks [26]; (c) restored saltmarshes of different ages as a space-for-time substitution to estimate the rate of carbon accumulation [27]; (d) dating downcore profiles using radionuclide ($^{210}$Pb) to determine sediment and carbon accumulation rates [28, 29]; and (e) repeat measurements of the elevation of sediment surface to quantify sediment deposition rates [30]. While all approaches highlight the potential for saltmarsh restoration to lead to carbon accumulation, each has limitations when used in isolation. A further challenge is that previous studies have either assessed only total carbon (which does not distinguish organic carbon from inorganic carbon such as

biogenic or lithogenic carbonates), or have quantified organic carbon using loss on ignition, which is known to have poor accuracy and large uncertainties [31].

There are a number of further considerations that could influence the net carbon benefit of a saltmarsh restoration or creation project, including changes in gas fluxes following tidal restoration [32]. One potentially important consideration is the balance between the carbon costs of constructing the site (e.g. building new flood defences inland and breaching the existing embankments, termed "managed realignment") and the carbon accumulation provided by the site [e.g. 33]. If project carbon costs are high relative to the rate of carbon accumulation, it may take years for the site to pay off the carbon debt of construction [34].

This research combines multiple techniques to evaluate carbon costs and benefits from saltmarsh creation through managed realignment. Over the course of several annual cycles we use remote sensing, field measurements and laboratory analysis of sediment to quantify total and organic carbon accumulation in an evolving saltmarsh in the first years after restoration. This allows us to reliably quantify the amount and rate of carbon accumulation following restoration. We then assess the carbon emissions incurred during site construction before identifying additional requirements for producing a full carbon budget for saltmarsh restoration.

## Materials and methods

### Study site

Steart Marshes (Somerset, UK; 51.20 N, 3.05 W) is a 250-ha managed realignment site, forming part of a larger 400 ha complex of restored wetland habitats managed by the Wildfowl and Wetlands Trust. It was constructed to create new intertidal habitat in compensation for previous losses, and to provide enhanced flood defences [35]. Prior to site construction, the land was under a mix of agricultural uses, including permanent pasture (i.e. pasture had been the land use over many years), grass ley (part of cyclical arable land management) and arable (winter wheat, barley, oilseed rape and maize) (Fig 1A). The site lies near the mouth of the River Parrett which drains a catchment of interbedded limestone and mudstone [36] and flows into the Severn Estuary. Hydrodynamic processes in the Parrett are dominated by a large tidal range which gives rise to strong tidal flows and large intertidal areas. At Hinkley, just to the

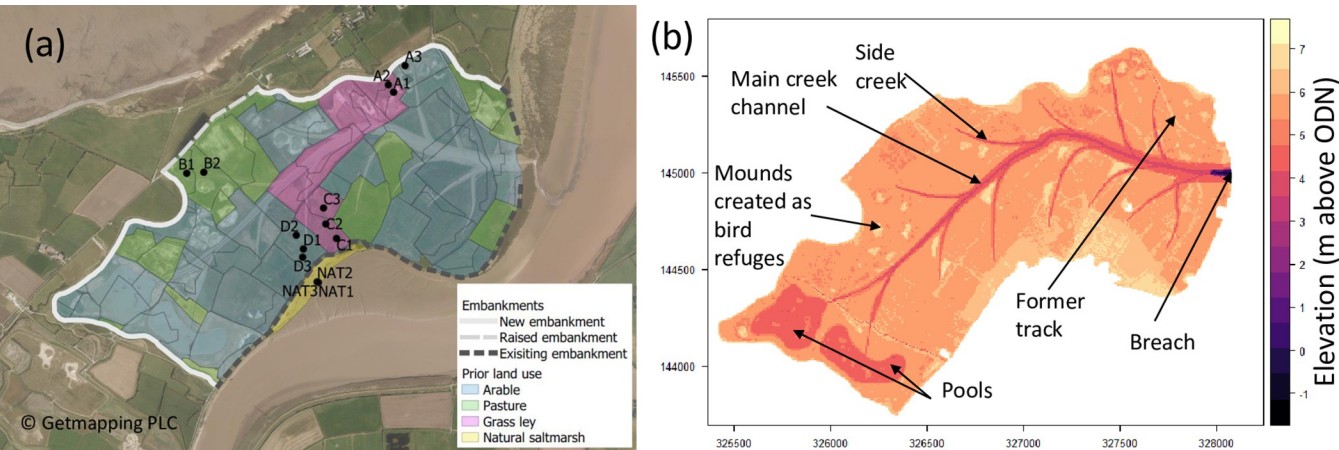

**Fig 1. Design and construction elements of Steart managed realignment, Somerset, UK.** a) Land use prior to the start of site construction in 2012, and locations of sampling points and the flood embankments constructed (new) or modified (raised) during the project; existing embankments that remained after the project are also shown. Land use classes created by authors based on data from Centre for Ecology and Hydrology Land Cover Map 2007 [82] and the project environmental statement [83]. b) Elevations across the site showing design and location of creek network, lagoons and islands. The location of the breach is also shown. Elevations based on LiDAR data from October 2014 [43].

west of the Parrett Estuary mouth, the mean spring tides have a high water height of 5.6 m Ordinance Datum Newlyn (ODN) and a low water height of -5.1 mODN, giving a range of approximately 11m [37, 38].

The construction of the managed realignment site started in early 2012, comprising the excavation of a creek network and pools, the construction of new flood defence embankments and the raising of a small length of existing embankment. The creek network (7.6 km total length) was designed to meet the geomorphological requirements of the scheme (see [39] for details), aid establishment of intertidal habitat, and minimise material transport distances by enabling construction of the required embankments from the excavated material [35]. In total, 4.75 km of new 4 m high or raised flood defence embankments were constructed (Fig 1A). All material used in the construction of the new embankments was obtained from the site, i.e. embankments were created from clays excavated from within site and no concrete was used in embankment construction. Several lagoons were excavated to enhance habitat provision for birds and fish, and islands were created from excess material to provide protected roosting and nesting locations for birds at elevations high enough to avoid excessive inundation by the tide [35]. In total, 489,422 $m^3$ of material was excavated and moved within the site during construction. A single, 250 m wide breach in the sea wall was created in September 2014, allowing regular tidal inundation to occur (further details of the breach are provided in [37]).

## Field sampling design

Four areas of the restoration site were selected for regular sampling, first an area substantially disturbed by earth moving vehicles during construction (Site A, Fig 1A) and three sites based on prior land use, permanent pasture (Site B), grass ley (Site C) and arable (Site D). Within these areas, we selected three sampling locations, stratified by the elevation prior to restoration of tidal inundation; the area of permanent pasture was relatively homogenous in elevation and so we only selected two sampling sites. To act as a natural reference, we selected a neighbouring area of pioneer saltmarsh (mostly bare ground with some *Spartina anglica*) and an area of saltmarsh with plant communities similar to those anticipated to establish on the managed realignment site, i.e. those dominated by *Puccinellia maritima* and *Aster tripolium* (NAT, Fig 1A). This gave a total of thirteen regular sampling locations within five sampling sites.

## Sediment collection, preparation, and storage

Sediments were sampled at each location immediately prior to restoration (28 August 2014, Sites A-D but not natural marsh), in December 2014 and then once or twice annually in 2015, 2016 and 2017, giving one pre-restoration and six post-restoration sampling time points (see S1 Table in S1 File for full details). Cores of 30-50cm were collected using a gouge soil auger (Royal Eijkelkamp, The Netherlands) and sectioned into 5–10 cm lengths for later analysis. Depending on site conditions, surface sediments deposited post-breach were sometimes difficult to sample using a soil auger as they were either prone to compression or highly friable, resulting in sample loss. In cases of minimal core compression (<2 cm), we retained the core (applying a compaction correction factor to the surface 10 cm [40], where this reflected the auger insertion depth at which the compression had occurred) and collected additional uncompressed surface samples using adapted syringe tubes. In dry condition when extensive mud cracking had occurred (see S1 Fig in S1 File 2015 images), coring resulted in significant sample loss. In these cases, the initial core was discarded, and we separately collected and sectioned the sediment polygon formed by the mud cracks, the depth of the mud crack was recorded, and deeper sediments were sampled by taking a core between mud cracks (with core section depth corrected to depth from surface). In total, we collected 78 cores (and associated

surface profiles), resulting in 596 samples. The horizon between the deposited silts and the underlying agricultural soils was determined through visual inspection of the cores (prior to sub-sampling) and the depth (in core and from surface) was recorded. The horizon was readily identifiable through a change in colour and texture of the soils, and by the presence of remnant vegetation and roots. Samples with a defined volume of 5 cm$^3$ were taken from the above-horizon section of the core or directly from surface sediments for dry bulk density measurements [41].

All samples were stored at 4°C prior to analysis. Dry bulk density was determined by drying the samples of a known volume to a constant weight at 105°C. The remaining core samples were dried at 60°C, covered, in aluminium trays/glass jars for approximately 96 hours, then ground using a pestle and mortar to ensure a homogeneous sample for further analysis.

## Quantifying carbon content of the sediment

We quantified the total carbon content (TC) of all collected samples. Total carbon contents were measured on dried, ground sediment samples using elemental analysis (LECO CR-412 Carbon Analyser (LECO Corporation, MI, USA) and Vario EL Cube (Elementar, Germany) instruments). Certified Reference Materials (CRM) were analysed on both instruments and replicates of an internal standard (bulk sample of surface sediments collected from the site) were included in all instrument runs. The measured carbon content (weight percent) of the CRM on the LECO instrument was consistently higher than the certified value (Leco Soil Standard 502–062 (n = 42), measured %C = 2.12±0.02, certified %C = 2.01±0.03), while analysis of the CRM on the Elementar instrument showed excellent agreement (Elemental Microanalysis Ltd Soil Standard B2184 (n = 6), measured %C = 2.29±0.06, certified %C = 2.31±0.06). Accordingly, all LECO measurements were multiplied by a correction factor of 0.95 (2.01/ 2.12). The corrected analysis of the internal standard on the LECO instrument showed excellent agreement with the Elementar analysis (LECO (n = 83), %C = 2.74±0.10; Elementar (n = 18), %C = 2.68±0.11).

To quantify total organic carbon contents (TOC), we selected one core from each site (A-D and NAT) from the most recent sampling period; for the restored marsh (sites A-D), TOC was quantified on the newly accreted sediment (above horizon) only. Aliquots of the same samples analysed for TC underwent acid digestion to remove inorganic carbon. Excess 1N HCl was added, and the samples were placed on a hotplate for three hours at 80°C [42]. Following the acid digest, the supernatant liquor was decanted, and the samples rinsed 3 times with deionised water before being taken to dryness. This process removes calcium and magnesium carbonates (aragonite, calcite, and dolomite), along with other water- or acid-soluble minerals, whilst minimising loss of labile organic matter. TOC losses from acidification were not quantified but are expected to be minimal and, furthermore, any labile organic carbon lost during this process would be that fraction most susceptible to remineralisation and thus least likely to be stored in the long-term. As such, our analytical approach results in a conservative estimate of organic carbon content, consistent with carbon accounting principles. Decarbonated samples were analysed on the Elementar instrument. Sample mass was recorded before and after decarbonation, with TOC values corrected to original sample mass, and the ratio of TOC to TC was determined for each sample analysed.

## Quantifying sediment deposition and erosion

Multiple Digital Terrain Models (DTM) at 50 cm horizontal resolution were obtained for the site, derived from airborne LiDAR data [43]. The final pre-breach imagery, from 10 July 2014, pre-dates significant earth movement on site and is therefore unsuitable for use as a baseline. Instead, we have used the first post-breach imagery, from 31 October 2014 (57 days post

**Table 1. Sedimentation at Steart Marshes measured by comparing LiDAR DTMs to a baseline survey on 31 October 2014, 57 days after sea defences were breached.**

| Survey date | Days since breach | Sedimentation rate (m yr⁻¹) | | Mean sediment depth (m) | Cumulative sediment volume (m³) | Cumulative carbon (95% confidence intervals) (t) | Cumulative organic carbon (95% confidence intervals) (t) |
|---|---|---|---|---|---|---|---|
| | | since start* | since previous survey | | | | |
| 24/01/2015 | 142 | 0.449 | 0.449 | 0.104 | 255646 | 12338 (2620–25697) | 6533 (1308–14101) |
| 05/04/2015 | 213 | 0.258 | 0.029 | 0.110 | 269282 | 13050 (3178–26582) | 6910 (1617–14704) |
| 04/06/2015 | 273 | 0.217 | 0.112 | 0.128 | 314126 | 15186 (4750–29578) | 8041 (2384–16392) |
| 31/07/2015 | 330 | 0.127 | -0.216 | 0.095 | 231555 | 11271 (1708–24269) | 5968 (875–13455) |
| 28/09/2015 | 389 | 0.153 | 0.277 | 0.139 | 341190 | 16491 (5625–31303) | 8732 (2837–17431) |
| 07/04/2016 | 581 | 0.110 | 0.034 | 0.157 | 384983 | 18626 (7016–34395) | 9863 (3536–19060) |
| 02/03/2017 | 910 | 0.092 | 0.064 | 0.215 | 525227 | 25507 (11315–44184) | 13506 (5614–24670) |
| 06/10/2017 | 1128 | 0.067 | -0.028 | 0.198 | 483641 | 23490 (10045–41212) | 12438 (5004–23134) |
| 01/04/2018 | 1305 | 0.073 | 0.105 | 0.249 | 608657 | 29541 (13545–49886) | 15642 (6733–28069) |
| 13/09/2018 | 1470 | 0.075 | 0.096 | 0.292 | 714513 | 34642 (16398–57400) | 18343 (8090–32402) |

breach), as a baseline for sediment accumulation on site. Between the date of the breach (4 September 2014) and the date of the imagery used, approximately 37 tides overtopped the creek banks and flooded some of the marsh surface (tides greater than 5.7 m ODN at the nearest available tide gauge, Hinkley Point (data from UK National Tide Gauge Network)). We obtained LiDAR DTMs for eleven further time points after breaching (see Table 1). Downloaded DTMs were processed in Rv4.02 [44] using the "raster" package [45]. Tiles were merged before being clipped by the restored site area. The site area was defined by manually drawing a polygon around the crest of the flood embankment to remove areas outside of the site. We then restricted analyses to locations subject to tidal inundation which were taken to be those areas below 7.07 m ODN, which is the level of the highest astronomical tides at the nearest port, Burnham-on-Sea [38]. The first DTM available after the breach (31 October 2014) was clipped to locations below 7.07 m and the resulting polygon (with an area of 244.7 ha) used to clip the remaining DTMs. In addition, two polygons were created on natural marsh areas to the north and south of the breach (S2 Fig in S1 File) and the elevation change between 2014 and 2018 LiDAR imagery was assessed.

Filtered DTM data should represent the ground elevations, but filtering does not completely remove dense, relatively short vegetation. Vegetation cover at the site in the first three years was sparse (S1 Fig in S1 File, H Mossman pers. obs.) and so we do not consider this an issue for those years; in the latest year, vegetation cover was denser and extensive, but unvegetated areas remained (H Mossman pers. obs.). The 50 cm resolution cannot account for surface morphology smaller than this (e.g. surface desiccation cracking, which was observed during summer months). We also observed sediment dewatering and shrinkage during dry periods, but these changes were small compared to interannual changes in elevation (Table 1). Pontee and Serato [37] quantified the variation in elevation of control points between years (the same LiDAR datasets we use) and found a mean vertical error of ±0.04 m.

Changes in elevation were calculated between each time point by subtracting the DTM of the first time point from the DTM of the more recent time point, and changes across the site visualised. Cumulative changes in elevation were calculated relative to the first post-breach DTM (31 October 2014, 57 days after breach). We calculated the mean elevation change across raster pixels, which was then converted to total change in sediment volume by multiplying by the area covered by the raster DTM. As an alternative way of visualising elevation change in the site, cumulative trajectories of elevation change were calculated for a random subset of 10,000 pixels.

To validate the elevation change obtained from LiDAR, we also conducted field measurements of elevation change. We measured elevation change *in situ* at one location within each area (Site A-D and NAT) using a modified sediment erosion bar [46], where one permanent 1.5 m metal stake was buried to a depth of 1 m, from which a portable 50 cm horizontal bar and supporting stake was established on a fixed compass bearing. The bar had 10 pins and the distance from the tip of the pin to the bar was measured. Stakes were installed on 14 and 15 December 2014, 14 weeks after the breach, and removed at the end of the study 5–7 March 2017 (S1 Table in S1 File).

## Data analysis: Variation in sediment carbon

Variation in sediment total carbon content was assessed as a function of depth using locally weighted polynomial regression (loess function in R), fitted separately for above and below the agricultural soil-new sediment horizon. We assessed whether there was a difference in carbon content in the newly accreted sediment, natural sediment (pooling locations, time points and depths for both) and the pre-restoration soils from the four land uses using Anova with a Tukey HSD post hoc test. Post-restoration samples from at or below the agricultural horizon were not included in this analysis because (1) we were interested in the carbon accumulating after the restoration in the newly accreted sediment and (2) elevated carbon contents were observed due to the burial of remnant agricultural vegetation as opposed to saltmarsh processes. The TC content of new sediment did not vary with depth in cores (r = 0.022, df = 144, P = 0.789) and there was no difference in the TC of newly accreted sediment (surface sample of sediment in each year) between years ($F_{1,46}$ = 0.369, P = 0.547). We therefore considered it justified to treat the carbon content of new sediment as coming from a single population (i.e. not varying between years). Thus, we calculated the mean and standard deviation of the TC content in newly accreted sediment, and also calculated the mean and standard deviation of the ratio of TOC to TC.

## Data analysis: Site-level carbon accumulation

Site-level carbon accumulation was determined over the full depth of the sediment accreted after the site breaching by (1) multiplying mean change in elevation (m, from DTMs) by site area ($m^2$) to obtain sediment volume ($m^3$), (2) multiplying this by bulk density of newly laid sediment ($t.m^{-3}$) to obtain sediment mass (t), and (3) multiplying this by sediment carbon content (%/100) to obtain total carbon accumulation (t). This was divided by site area to obtain $tC.ha^{-1}$. This calculation was repeated with the additional step of multiplying by the ratio of TOC to TC to estimate site level total organic carbon accumulation.

As each stage in this calculation involves measurements made with error, we used Monte-Carlo resampling to estimate site-level carbon accumulation while propagating errors from each step. If elevation measurement errors were independent for each DTM pixel in each time point then errors largely cancel out. A more conservative approach is to assume that measurement errors apply systematically to a survey. We do the latter, and take mean elevation change

between surveys as coming from a normal distribution with a mean equal to the measured change in elevation, and a standard deviation of 0.04 m based on measurements of control points [37]. Bulk density of newly accreted sediment was sampled from a normal distribution with mean 1.11 and SD 0.27 t.m$^{-3}$. Sediment TC was sampled from a normal distribution with mean 4.37% and SD 0.50%, and the ratio of TOC to TC was sampled from a normal distribution with mean 0.53 and SD 0.08. We took 100,000 samples from these distributions to obtain a distribution of carbon accumulation estimates.

## Carbon costs of construction

The carbon cost of constructing the wider 400 ha Steart Marshes complex (comprising the 250 ha managed realignment site and neighbouring areas of freshwater wetland) was estimated using the Environment Agency's basic carbon calculator (version 3.1.2, dated 2010 (unpublished); since incorporated within the e:Mission Eric carbon planning tool [47]), with the final estimate produced at the end of construction in January 2015. The calculator included estimated greenhouse gas emissions (in carbon dioxide equivalents, $CO_2$e) from fuel used for personnel travel, energy use on site within portable accommodation, the emissions embodied in construction materials (considering the weight of material and distance transported to site), and a first order estimate of emissions from machinery fuel usage.

As all the materials for the embankment construction were obtained within the footprint of the site, the principal source of emissions was the fuel consumed by construction machinery moving the material within the site. As such, we have refined the estimate of machinery fuel usage based on the known volume of earthworks undertaken for the whole scheme, where fuel consumption was estimated by considering the work required, fuel burn per hour and productivity per hour. Within the managed realignment site, the amount of material that was excavated and transported was calculated by considering the size of the creek network and the volume of the embankment. The material was excavated using an EC250DL Excavator, transported across the site in a Volvo A25D Articulated Dumper Truck (capacity 10.7 m$^3$) and constructed in situ with a D6 Bulldozer and Roller (S2 Table in S1 File). The distance travelled was calculated based on the distance from each section of the embankment to the nearest source of materials, and fuel burn and productivity were obtained from manufacturers and suppliers. Fuel consumption associated with earthworks in other areas of the Steart Marshes complex were based on the volume of earth moved and ground conditions in comparison to the managed realignment site. A potentially important construction impact we did not account for was the changes in the carbon stocks in the soil resulting from excavation, movement and reburial. This could result in losses of organic carbon as it becomes oxidised. For example, the repeated disturbance of ploughing when former pastures are converted to croplands results in a loss of carbon of 0.95 t.C.ha$^{-1}$.y$^{-1}$ [48]. However, losses during construction at Steart are likely to be lower as 1) material was excavated, quickly moved to the new sea wall and reburied; 2) the top soil comprises the top c.30 cm of soil so much of the material would be sub-soil that contains less carbon; 3) large areas of the site had land uses (arable and to a lesser extent grass ley) that involve repeated disturbance of the soil (Fig 1). Carbon accumulation potentially occurs on the sea wall following grass establishment, which may offset some of the losses, and this was also not accounted for.

## Results

### Sedimentation rates

Field measurements of sedimentation between December 2014 and March 2017 indicated a mean rate of 0.048 ± 0.013 SE m yr$^{-1}$ on the restored site. As these sampling points were only

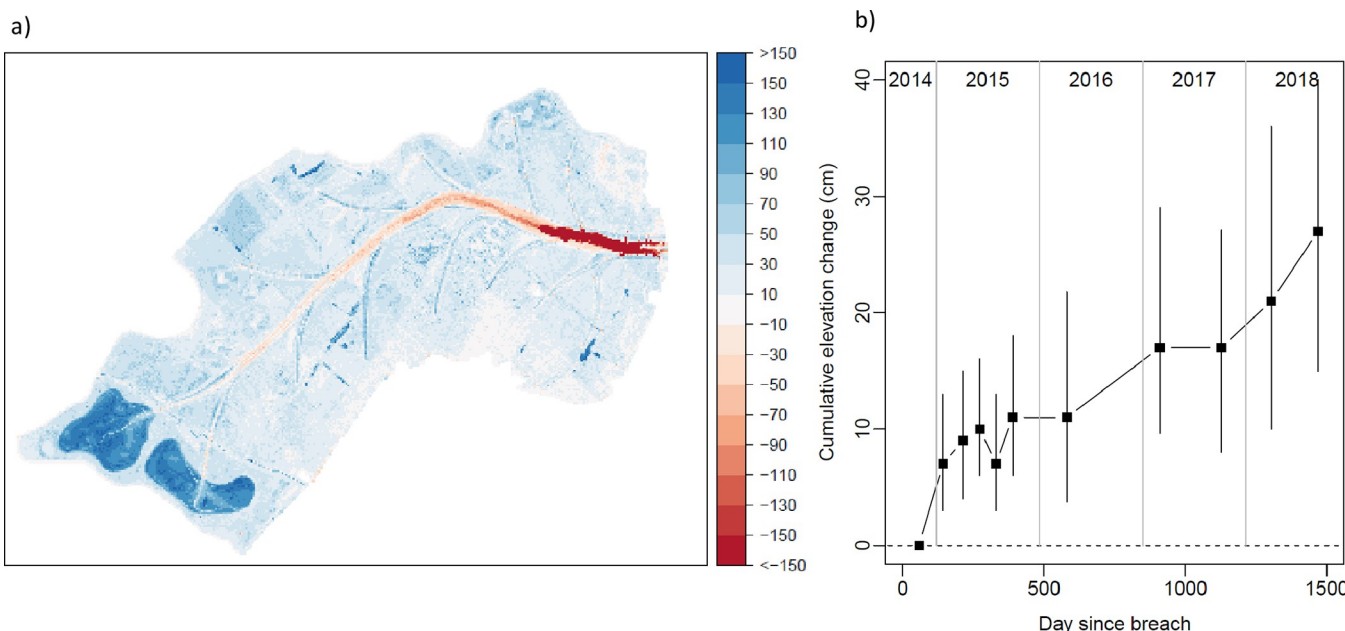

**Fig 2. Cumulative sedimentation at Steart Marshes calculated from LiDAR DTMs (LiDAR data obtained from data.gov.uk).** (a) Change in elevation (cm) between 13/09/2018 (1470 days since breach) and 31/10/2014 (57 days since breach). (b) Cumulative change in elevation over time for individual 50x50 cm pixels. Points show median cumulative change for a random sample of 10,000 pixels. Error bars show the interquartile range for the same sample of pixels.

located in a small portion of the site, comparison of successive DTMs were used to assess sedimentation across the site. These indicated that the net elevation of the site increased over time (Fig 2), and by September 2018 714,513 m$^3$ of sediment had accumulated across the site, with an average depth of 0.292 m and sedimentation rate of 0.075 m yr$^{-1}$ (Table 1). DTM derived elevation change was closely related to field measurements of elevation change (Fig 3), although sampling periods were not exactly comparable. DTM elevation change between October 2014 and March 2017 was strongly related to field measurements but biased towards higher sedimentation for DTMs as these covered a longer period ($\Delta$DTM$_{Oct14-Mar17}$ = 71.3 + 1.04 $\Delta$Field$_{Dec14-Mar17}$, R$^2$ = 0.775, F$_{1,9}$ = 31.1, P < 0.001, note coefficients in units of mm). DTM elevation change between January 2015 and March 2017 was also strongly related to field measurements, with no systematic bias over the range of observed sedimentation values ($\Delta$DTM$_{Jan15-Mar17}$ = 29.7 + 0.74 $\Delta$Field$_{Dec14-Mar17}$, R$^2$ = 0.686, F$_{1,9}$ = 19.6, P = 0.002, Fig 3). *In situ* measurements of elevation change on the natural marsh found an increase of 0.26 m in total between December 2014 and March 2017. LiDAR assessment of elevation change on the natural marsh south of the breach was 0.281 m and -0.003 m to the north of the breach between 2014 and 2018.

There was no clear trend in DTM-derived sedimentation rate with time since breach (regression: slope < 0.001, F$_{1,8}$ = 0.35, P = 0.568), although the most rapid sedimentation was noted immediately following the breach (Table 1). The net elevation of the site increased between most LiDAR surveys. However, in two instances mean elevation decreased between consecutive LiDAR surveys (between June and July 2015, and between March and October 2017), indicating reduction in sediment volume most likely due to dewatering over the summer months.

Within the site, elevation change varied from net accretion of 2.2 m to net erosion of 5.0 m (Fig 2A), with 92% of DTM pixels experiencing net accretion and 7% experiencing net erosion. Some locations experienced considerable erosion (S3 Fig in S1 File), especially in the main

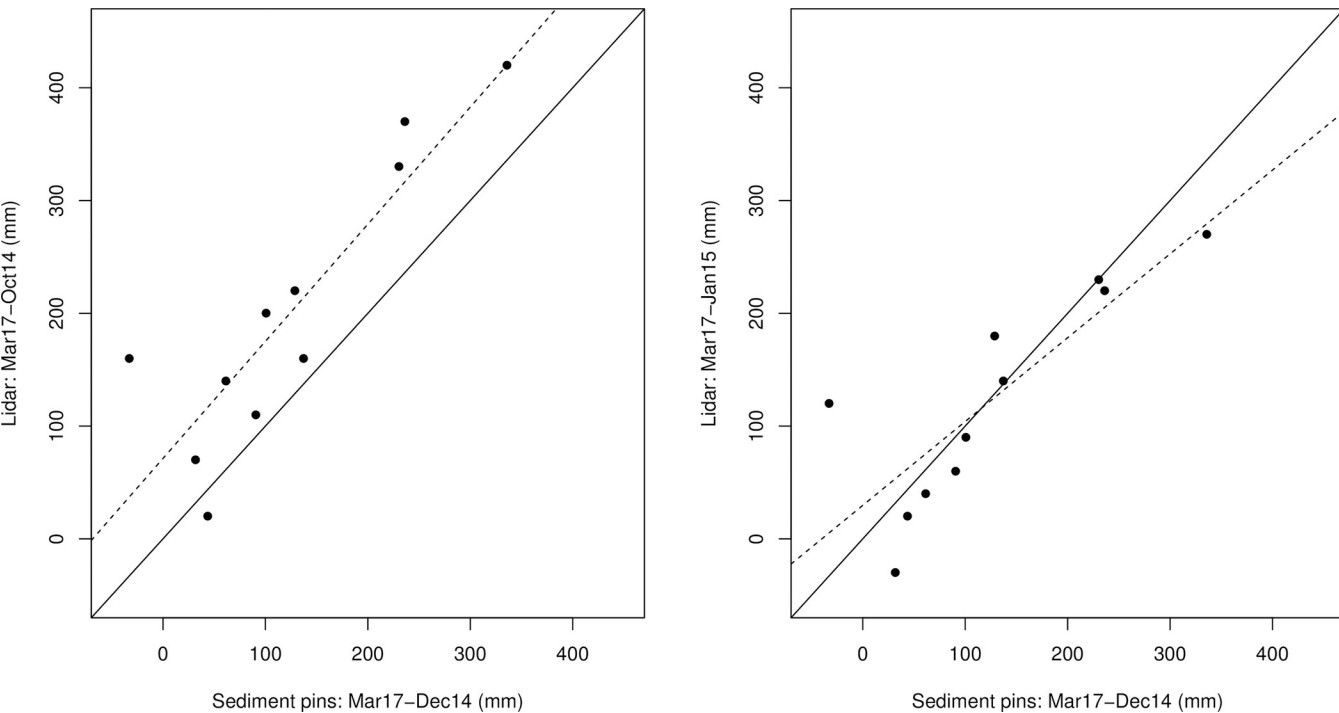

**Fig 3. Relationship between elevation change measured with LiDAR derived-DTMs and in situ measurements with pins.** In situ measured data (x axis) show difference in elevation between December 2014 (3 months after restoration) and March 2017. Left: Compares in situ data to elevation changes derived from LiDAR data taken in October 2014 and March 2017, and Right compares elevation changes between January 2015 and March 2017. No LiDAR images are available for December 2014. Solid lines show a 1:1 relationship and the dashed lines show the actual relationship (linear regression) between DTM-derived and in situ measurements (dash lines Left: $R^2$ = 0.775, P <0.001; Right $R^2$ = 0.686, P = 0.002). LiDAR measurements are strongly related to in situ measurements and are not systematically biased when sampling periods are more closely matched (i.e. Right).

creek which deepened progressively in an upstream direction over time (S4 Fig in S1 File, see [37] for analysis of the main creek profile). Away from the main creek, most locations increased in elevation. This increase was most evident in the excavated pools at the rear of the site, and to a lesser extent in the side creeks (Fig 2A); 558,648 out of 9,792,179 pixels experienced > 1m of accretion, and 92% of these were located in the two pools. Elevation change was also related to initial elevation (generalised additive model, effective df = 8.8, F = 369.2, P < 0.001, deviance explained = 25%), with slower sedimentation rates at higher elevations (Fig 4). Variation in elevation change was most constrained at higher elevations, while at lower elevations (< 5 m ODN) some locations experienced marked accretion and others experienced marked erosion (Fig 4).

## Properties of newly accreted sediment

The TC content of newly accreted sediment was significantly different from both the natural saltmarsh and the pre-restoration soils ($F_{5,249}$ = 48.7, p<0.001, Fig 5). Soils collected prior to restoration from all land uses had significantly lower TC contents than the newly accreted sediment and the natural saltmarsh sediments, with those from the pre-restoration disturbed (A) and arable (D) areas having the lowest carbon contents (Fig 3). Sediments from the natural saltmarsh had significantly higher TC (4.72 ± 0.58%) than the newly accreting sediment on the restoration site (4.37 ± 0.50%). The ratio of TOC to TC was similar in natural saltmarsh and newly accreting sediment (natural = 0.524, restored = 0.529, Fig 5), giving a TOC of 2.48 ± 0.40% on the natural saltmarsh and 2.31 ± 0.44% in newly accreted sediment on the

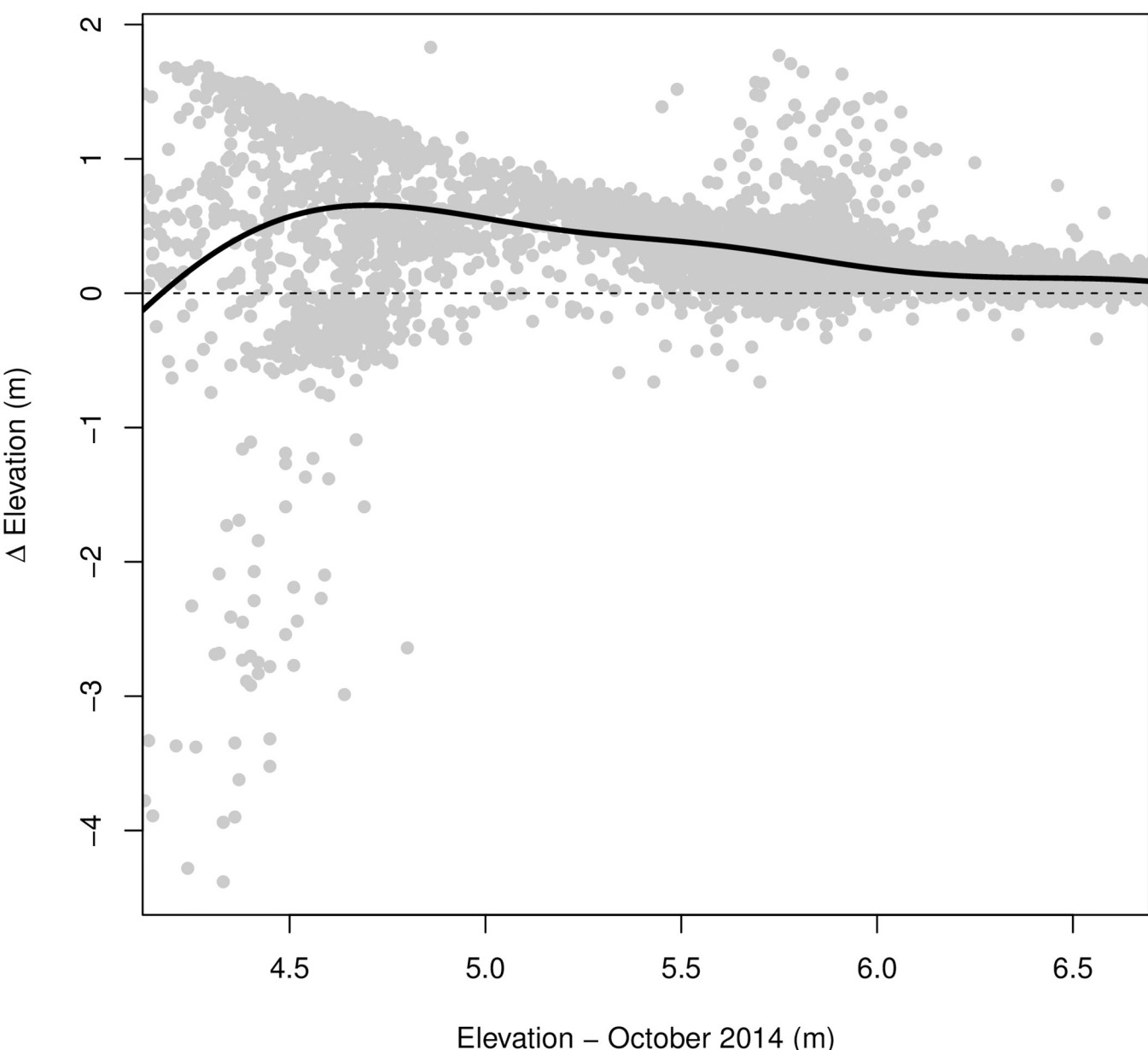

**Fig 4. Relationship between elevation change (2014–2018) and initial elevation for a random sample of 10,000 pixels taken across Steart Marshes.** The solid line shows the relationship between elevation change and starting elevation modelled by a generalised additive model (effective df = 8.8, F = 369.2, P < 0.001). For clarity, the x-axis limits have been clipped to show only the middle 95% of data (4.25–6.60 m starting elevation), but data from all elevations were used to fit the generalised additive model. The dashed line indicates an elevation change of zero (i.e. no net accretion or erosion).

restoration site. The bulk density of newly accreted sediment ranged from 0.553 to 1.568 t m$^{-3}$ (mean = 1.110 ± 0.267 SD), with no systematic differences between sampling locations ($F_{4,25}$ = 0.924, P = 0.466). The carbon density of newly accreted sediment on the restored saltmarsh thus on average contains 0.049 t m$^{-3}$ TC and 0.025 t m$^{-3}$ TOC.

There was some spatial variation in the TC content of new sediment between sampling sites ($F_{3,44}$ = 5.1, P = 0.004), with significantly lower TC contents in the disturbed site than the arable site (Tukey post-hoc test, difference = 0.68, P = 0.002). There was a non-significant tendency for TC content to be lower at higher elevations (Spearman's rank correlation between

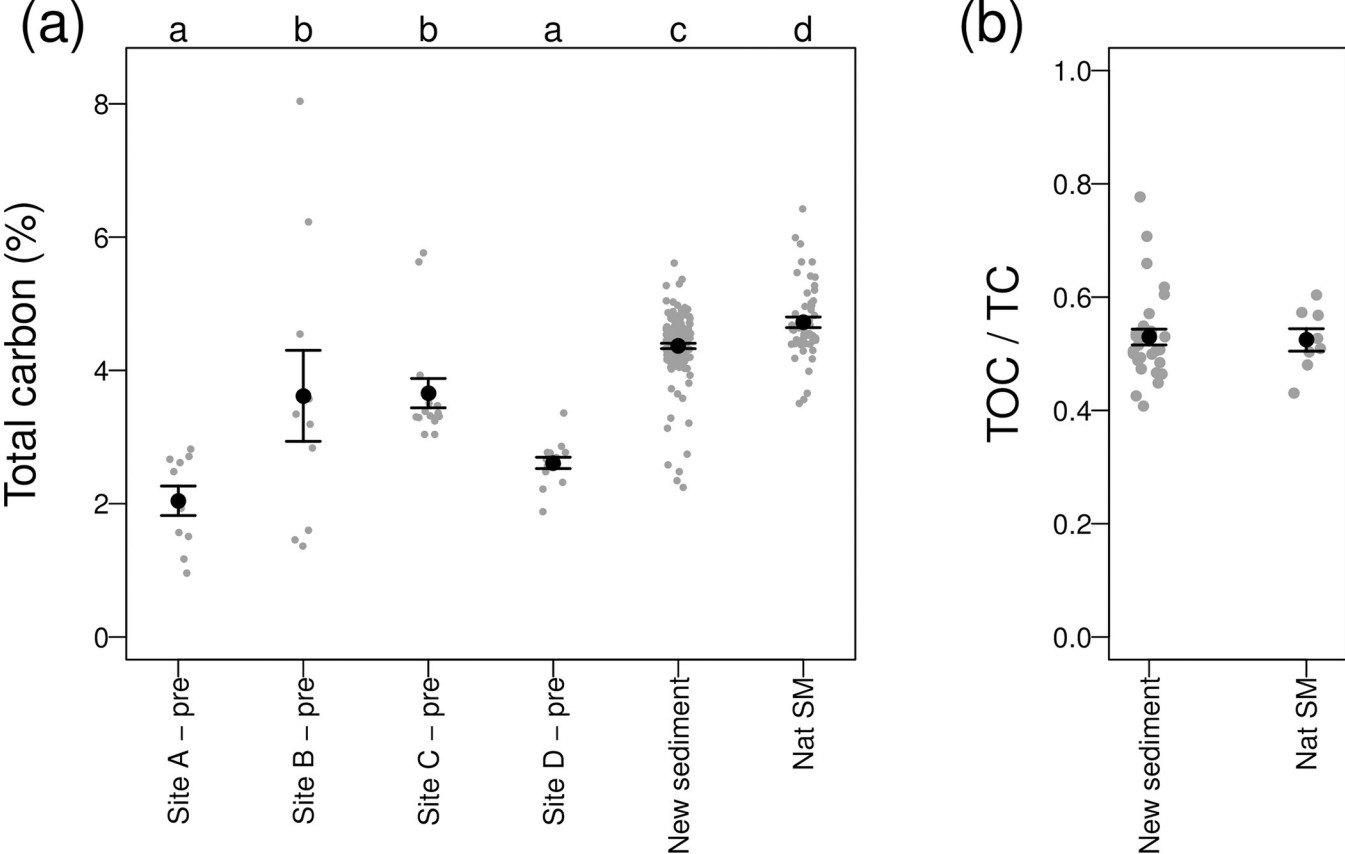

**Fig 5. Total carbon content of soil and sediment samples collected from Steart Marshes before and after the restoration of tidal inundation.** Soil samples were collected prior to restoration from an area heavily disturbed during construction (site A), an area of pasture (site B), grass ley (site C) and arable (site D). 'New sediment' are samples of newly accumulated sediments from the restored site after restoration, with data from all locations and time points pooled. Sediment was also collected from an adjacent natural saltmarsh. Differing letters denote significant differences in the carbon content of sediments between locations (P < 0.05).

starting elevation and mean TC for each sampling location, $r_s$ = -0.527, P = 0.100). The ratio of TOC to TC also differed between sampling sites ($F_{3,27}$ = 5.41, P = 0.005), with a higher ratio of TOC to TC in the permanent pasture site than the grass ley and disturbed sites (Tukey post-hoc test, P = 0.005 and P = 0.046 respectively). There was no significant change in TC content over time (relationship with year, $F_{1,46}$ = 0.369, P = 0.547); TOC measurements were only made for a single year so we cannot explicitly assess if this changed. The lack of change over time in TC content was supported by it not changing with depth (Fig 6). Some samples taken at the horizon with the underlying agricultural soils had very high carbon content, reflecting the terrestrial vegetation burried by the initial inundations of sediment. Below the horizon, TC was lower than in newly accreted sediment and directly comparable to the pre-breach measurements of the agricultural soils (Fig 6).

## Carbon balance

Between 31 October 2014 and 13 September 2018 714,513 m$^3$ sediment accumulated on the site. Based on the measured properties of this sediment (mean bulk density of 1.110 ± 0.267 t m$^{-3}$ SD, TC of 4.367 ± 0.499%) this equates to 34,642 tC (95% confidence intervals = 16,398–57,400) accumulated in sediment, at a rate of 36.6 t C.ha$^{-1}$.yr$^{-1}$ (95% CI = 17.3–60.6).

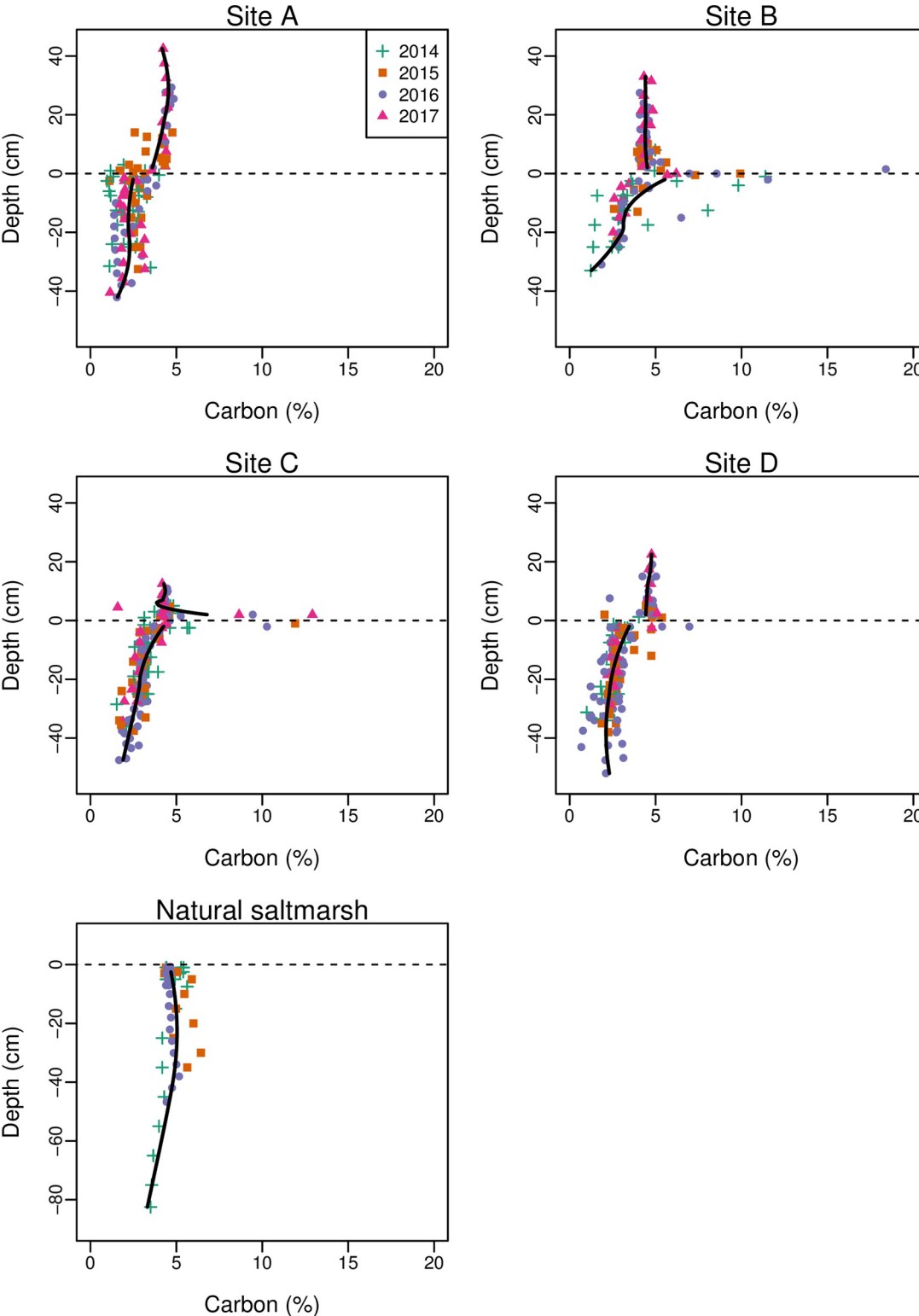

**Fig 6. Relationship between soil carbon content and depth.** Cores were taken each year at three locations in each starting land-use. Depths are expressed relative to the horizon between agricultural soil and newly deposited sediment, except for natural saltmarsh where depths are from the surface (note difference in y-axis scale for natural saltmarsh). Lines show fits of locally weighted polynomial (loess) models pooling data across locations and years. Loess models have been fit to new sediment (depth > 2 cm) and old sediment (depth < -2 cm) to reduce the effect of vegetation on the horizon.

Restricting this to TOC (53.0 ± 7.8% of TC) gives 18,343 tC (95% CI = 8090–32402) accumulating at 19.4 tC.ha$^{-1}$.yr$^{-1}$ (95% CI = 8.5–34.2).

We estimated the carbon costs of site construction in order to compare this to the carbon accumulation of the site. In total, 489,422 m$^3$ of material were excavated on site, with 411,397 m$^3$ used in the construction of the new flood embankments and the remainder used in site landscaping. Moving material across the site resulted in vehicles travelling 69,563 km. Overall, construction of the managed realignment site earthworks required 551,012 litres of diesel fuel to be combusted, resulting in 1,477 tCO$_2$e (403 tC) being emitted (S2 Table in S1 File). An estimated additional 20% of fuel consumption was assumed for the construction of earthworks in other areas of the Steart Marshes complex, giving total emissions associated with machinery fuel usage of 1,772 tCO$_2$e (483 tC). Combining these figures with the estimated emissions from personnel travel, energy use in portable accomodation, and embodied emissions of construction materials (including rammed earth to construct the embankment) from the Environment Agency carbon calculator, gives estimated total construction emissions of 2,762 tCO$_2$e (753 tC).

## Discussion

We find that Steart Marshes managed realignment has rapidly accumulated carbon since the fronting flood defence embankment was breached, and that this carbon accumulation is 50 times greater than the estimated direct carbon costs incurred during site construction. The rate of carbon accumulation at Steart Marshes (TC = 36.6 t C.ha$^{-1}$.yr$^{-1}$, TOC = 19.4 t C.ha$^{-1}$.yr$^{-1}$) is considerably higher than has been found at other sites. In the Bay of Fundy, which like the Severn Estuary is hypertidal, carbon accumulation is lower but within the same order of magnitude at 13.29 t C ha$^{-1}$ yr$^{-1}$ [30], but rates at other sites are an order of magnitude lower than at Steart Marshes. For example, saltmarshes in eastern England were reported to accumulate carbon at a rate of 1.04 t C ha$^{-1}$ yr$^{-1}$ for the first 20 years following creation [27], while a recovering saltmarsh in Australia accumulates at a rate of 0.5 t C ha$^{-1}$ yr$^{-1}$ [49]. The rate of carbon accumulation in a restored saltmarsh is a product of the rate of sediment accumulation and the carbon density of that sediment, and we can look at both these elements to see if Steart Marshes is unusual compared to other restored sites.

Steart Marshes has experienced rapid sediment accumulation since it was breached (mean rate of increase in elevation = 75 mm yr$^{-1}$, Table 1). Similarly high accretion rates have been reported from elsewhere in the Severn Estuary system (short-term accretion rates of 60mm yr$^{-1}$ in young marshes in Bridgwater Bay [50]; around 60mm yr$^{-1}$ in accreting natural marsh in Portishead [51, 52]) and also in the Bay of Fundy (>60 mm yr$^{-1}$ [30]). In comparison, reported sedimentation rates for Tollesbury and Freiston Shore managed realignment sites in eastern England are considerably lower (< 20 mm yr$^{-1}$ [53–55]). A recent meta-analysis has indicated that sediment availability is the dominant control on the vertical accretion of coastal wetland restoration projects, with tidal range, elevation within the tidal frame and sea-level rise explaining a smaller amount of observed variation in the vertical accretion rates of saltmarshes [56]. Hypertidal systems such as the Severn Estuary and the Bay of Fundy are characterised by very high energies and dynamic intertidal sedimentation, where the high suspended sediment load (due to the turbulence created by tidal currents and bores) allows deposition during both flood and ebb tides [57]. While the suspended sediment concentrations within the Severn Estuary vary significantly (depending on geographical location, position in the water column, and state of tide), there is a turbidity maximum located in the lower estuary in the vicinity of Bridgewater Bay and the Parrett Estuary (and thus Steart Marshes), with high suspended sediment concentrations typically in the range of 1,000–10,000 mg/l with values often exceeding

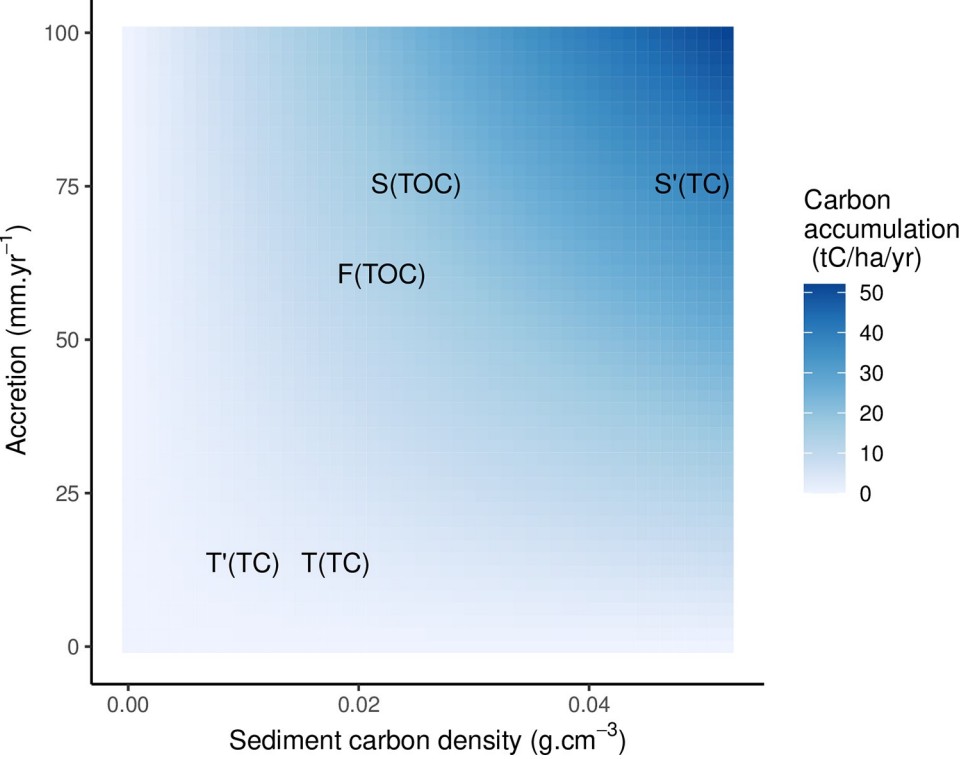

**Fig 7. Carbon accumulation potential (tC ha$^{-1}$ yr$^{-1}$) of saltmarsh restored by managed realignment.** The coloured surface shows rates of carbon accumulation for each combination of accretion and carbon density. Observed values from Steart (S and S', this study) and published studies at Tollesbury (T [high marsh] and T' [low marsh] from [54, 84]) and the Bay of Fundy (F, [30]) are mapped on to this carbon accumulation space. TC indicates total carbon density, and TOC indicates total organic carbon density.

100,000 mg/l [58–61]. Much lower suspended sediment concentrations (~50–150 mg/l), and thus lower sediment supply, are reported for the Blackwater Estuary (Tollesbury) and The Wash (Freiston Shore) [62, 63].

Sediment bulk density at Steart Marshes was 1.1 t m$^{-3}$, which combined with an organic carbon content of 2.2% gives a carbon density of 0.025 t m$^{-3}$. Comparison with values from other managed realignments indicates bulk density varies from 0.74–1.4 t m$^{-3}$ [27, 64–66] and carbon content varies from 1.8–4.23% (range includes total carbon and TOC values; cf TC 4.4% and TOC 2.2% in this study). Combining all combinations of sediment carbon content and accretion rates gives the space of potential carbon accumulation rates in saltmarsh restored by managed realignment (Fig 7). This indicates that Steart Marshes has high rates of carbon accumulation because it experiences both high rates of accretion and has relatively high sediment carbon density; thus while neither variable is exceptionally high compared with other values reported in the literature, this combination leads to the exceptionally high rates of carbon accumulation. Lower values of either one of these limits carbon accumulation. For example, In natural saltmarshes in China, carbon accumulation rates are low (0.35–3.61 tC ha$^{-1}$ yr$^{-1}$) despite accretion of 20 mm yr$^{-1}$ because of low sediment carbon densities ($< 0.01$ t m$^{-3}$) [67]. This variability in carbon accumulation rates between sites highlights the need for further work to support large-scale assessments of the carbon accumulation potential of saltmarsh restoration. For example, TOC accumulation rates at Steart Marshes are over 18 times higher (and TC accumulation rates are over 35 times higher) than those used in a recent study to estimate the UK's carbon accumulation potential [68], while Mossman et al. [34] found ~13 fold

variation in the potential amount of carbon accumulated by restored saltmarshes in the UK based on published estimates of carbon accumulation.

Our analysis assumes that soil properties (soil carbon, bulk density) come from a single statistical population across the site and over time. However, there were small differences in the carbon content of new sediment across the site, with differences between prior-land classes and a non-signficant tendancy for the proportion of carbon to be lower at higher elevations. The reasons for this are unclear, but could relate to spatial variation in algal films and vegetation establishment across the site. However, total carbon content did not increase with time as the site became more vegetated (we lack data to assess this for organic carbon). If the drivers of variation in sediment carbon across the site were known this could be used to scale-up and refine estimates, but this is not currently possible. Our treatment of carbon content as a single population is also supported by the lack of change in carbon content with depth, indicating that newly deposited sediment has a similar carbon content over time. Bulk density was averaged from samples taken near the surface, which ensured we captured newly accreted sediment, but may bias towards underestimating average bulk density due to compaction of sediment with depth. Not accounting for these potential changes with depth would mean our carbon density (and total carbon accumulation) estimates are likely to be conservative. The bulk density of sediment would be expected to exhibit temporal variation, with lower bulk density (but greater sediment volume) when sediment is waterlogged (e.g. winter, spring), and higher bulk density (but lower sediment volume) when sediment is dry (e.g. summer, early autumn). Our bulk density measurements come from spring and summer, so should capture this temporal variation in bulk density. However, explicitly quantifying temporal variation in bulk density would allow temporal coupling with sediment accumulation data and thus refined quantification of intra-annual variation in carbon accumulation–apparent reductions in carbon stocks over the summer when sediment volume reduced may not occur in reality because of a concurrent increase in sediment bulk density.

## Future changes in carbon accumulation

Although we found the fastest rates of accretion shortly following breaching, we did not find a statistically significant reduction in accretion rates over time. However, a reduction in accretion rates would be expected as the saltmarsh develops. This is because accretion rates tend to be faster at lower elevations which experience more frequent tidal inundation (Fig 4, [69]), and as these lower areas increase in elevation they experience fewer inundations, and thus slower accretion. Indeed, space-for-time substitutions indicate that carbon accumulation rates slow over time [27]. It is likely that carbon accumulation rates at Steart Marshes would slow with longer monitoring. Assuming there is sufficient sediment available (as very likely in this case), accretion at managed realignments is expected to occur until the site is a level plane accreting in line with sea-level rise [69]. Natural saltmarsh surrounding the site occurs at elevations of 6.5 m, where if accretion at Steart Marshes stablised at this level this would result in a TC accumulation in excess of 100 ktC (TOC in excess of 50 ktC), of which 33% has currently been accumulated. Even after this point, saltmarshes can continue to accrete with sea-level rise assuming there is suffcient sediment [70], which at 3.7mm yr$^{-1}$ [71] would result in continued TC accumulation of 439 t C yr$^{-1}$ and TOC accumulation of 225 t C yr$^{-1}$ at Steart Marshes assuming no change in sediment carbon content.

## Challenges with determining the carbon budget of a managed realignment

Our results indicate that carbon accumulation at Steart Marshes exceeded direct construction costs. However, there are a number of uncertanties that would need to be considered to

produce a full quantive carbon budget and determine the net carbon benefit of the site (Table 2, [34]). Some assumptions, such as assuming the carbon content of sediment lost is the same as sediment gained, are likely to mean our estimate of carbon accumulation is conservative (Table 2). However, evealuation of other factors, such as greenhouse gas emissions from the site, or determing the fraction of autochthonous and allochthonous carbon, would offset some (if not most) of the observed carbon accumulation (Table 2). These are discussed further below.

**Table 2. Elements that require consideration in the quantification of a full carbon budget of a managed realignment site.** The aspects included in this study, the approaches to these that we took, and any implications of these approaches are also given.

| Element | Approach in this study | Rationale of approach and its implications |
|---|---|---|
| Amount of sediment gained and lost within the site | Measured using LiDAR derived DTMs, validated against *in situ* measurements. | Baseline LiDAR is 57 days after breach and ~37 tides covered at least some of the marsh surface between the breach and this LiDAR image, so some post-breach sedimentation will have been missed. This approach has likely underestimated total carbon gained. |
| Changes in carbon stored by intertidal habitat in the wider estuary as a result of the realignment site | Not considered in this study. | A five year monitoring programme for the scheme found no evidence that the scheme had caused increased erosion in the main estuary channel bed [85]. Near to the scheme, the most significant changes have been associated with the erosion of the exit channel due to the strong flows into and out of the realignment site [37]. Erosion in the exit channel has progressed into the site through the creation of a distinct step, and these changes within the site are captured in our analysis. |
| Carbon in sediment gained | Cores taken at 11 locations in the restored site, average carbon content of new sediment used. | We found total carbon and total organic carbon in new sediment was somewhat lower than in the adjacent natural saltmarsh. The site was in the early stages of restoration (first 4 y). Further development of biotic communities, particularly the vegetation, may increase the carbon content of the sediment at the restored site.<br>Some spatial variation in carbon contents of new sediment around the site. Reasons for this were not clear but understanding this would allow more spatially refined models of carbon accumulation could be made. |
| Carbon in sediment lost (eroded) | Assumed to be the same as carbon gained. | Carbon content of the soils eroded (e.g. from main creek) is likely to be lower than that in the new sediment because the agricultural soils significantly had lower carbon. However, this could not be quantified because erosion in the main creek was up to 5 m deep. Some of the erosion later in the study would have been of newly accreted sediments (e.g. due to formation of small creeks) and thus of same carbon content as that gained. In total, this approach has likely underestimated total carbon gained. |
| Source of carbon in sediment | Not considered | Burial of *in situ* derived carbon would be a true gain. Carbon from outside the site may have ended up being stored elsewhere in the absence of the site, or may have been oxidised; the extent to which either happens is uncertain. The total carbon accumulation here provides an upper bound for net carbon buried. |
| Plant biomass | Not considered | Belowground biomass contributes to carbon accumulation, and is expected to increase as the site became more vegetated. Similarly, more vegetation creates a source of carbon to be buried. |
| Greenhouse gas fluxes | Not considered | Release of greenhouse gases (e.g. methane) may offset some carbon accumulation, although increased salinity may reduce greenhouse gas release.<br>Burden et al. [84] found $CH_4$ and $N_2O$ fluxes were close to zero on a restored and natural saltmarsh in Essex. However, Adams, Andrew & Jickells [86] suggest gas fluxes could reduce carbon sequestration on MR sites by 24%. |

*(Continued)*

**Table 2.** (Continued)

| Element | Approach in this study | Rationale of approach and its implications |
|---|---|---|
| Construction carbon costs | Calculated using estimated fuel use during construction in combination with estimates from the Environment Agency's basic carbon calculator (version 3.1.2) for personnel travel, energy use in portable accommodation, and the embodied emissions associated with construction materials. | Creation of the embankments is very likely to be the greatest construction carbon cost of the managed realignment, and itself is small compared to the carbon gained by the habitat created on the site.<br>All material for the managed realignment part of the site were locally-won material for the embankments from borrow pits on site and was not imported.<br>Since the completion of the Steart project, a more up-to-date Environment Agency carbon estimation tool became available, which considers the carbon of other project stages (operational, decommissioning) to provide the whole-life carbon over a 100-years. The tool currently cannot be adjusted to deal with locally-won embankment material, and was not used here.<br>Some operational carbon cost will occur from the site managers WWT, but has not been included in the calculations. This would include activities associated with site inspections and the maintenance of the embankments (such as grazing by sheep rather than mowing). It is expected to be minimal compared with construction. No decommissioning is anticipated.<br>Changes in soil carbon due to excavation have the potential to be large to the amount of sediment moved. However, the majority would be sub-soil and so low in carbon. Movement and reburial was rapid, limiting oxidation. Disturbance-dominated land uses (i.e. arable) would have lowered soil carbon prior to restoration. |
| Prior land use–some changes in land use may result in substantial carbon lost, e.g. loss of trees | Not considered | The site was a mix of arable and pasture prior to restoration and relatively few trees were removed during construction, but this should be considered for future sites. |

One of the key elements of the carbon budget we have not quantified is the complex changes in the fluxes of greenhouse gas following saltmarsh restoration. Temporal and spatial variation in hydrology (particularly the position of the water table relative to the sediment surface) and salinity have significant effects on the biogeochemical drivers of $CH_4$ and $CO_2$ emissions [72]. Tidal inundation brings an influx of sulphate ions, which has the potential to inhibit microbial production of $CH_4$ [32], meaning that the restoration of tidal inundation to coastal wetlands has been modelled to lead to large reductions in $CH_4$ emmisions [73]. Estuarine salinity close to our site is thought to be >28 ppt throughout the year [74], and these polyhaline conditions are reported to have lower methane emissions than in lower salinities [75]. However, recent evidence shows that methanogenesis can co-occur with sulphate reduction within highly saline saltmarsh sediments, suggesting that an increase in marsh salinity caused by tidal re-connection may not necessarily inhibit the production of $CH_4$ by the soil microbial community [72]. This may explain why other studies have reported very high net $CH_4$ fluxes from some restored salt marshes despite observing high levels of local salinity [76]. Plant community composition and productivity can also affect greenhouse gas fluxes [77], and these effects may vary with vegetation succession on restored sites.

A particularly challenging element of quantifying the full carbon budget of a restored saltmarsh, and one we have not quantified, is determining (a) the nature and origin of the carbon accumulated in sediment (autochthonous or allochthonous) and (b) the fate of that carbon relative to it's fate in the absence of the restored site (preserved and sequestered in the long-term, oror decomposed and released back to the atmosphere). This is critical to establish the additionality test needed for carbon codes and offsetting credits (e.g. Verified Carbon Standard Methodology VM0033 [78]), where in simple terms, carbon credits should only be generated

through the creation of new net sinks of atmospheric carbon. While long term preservation of autochthonous carbon can be counted as a carbon benefit, how to account for allochthonous carbon is more complex. Concerns regarding additionality have led to variable treatment of allochthonous carbon because creating a new apparent store could be depleting supply to an adjacent system (i.e. the carbon would have been stored elsewhere in the absence of the project). In mineraogenic salt marshes, the majority of organic carbon is allochthonous in nature [79]. Indeed, in the early evolution of the mineraogenic Steart Marshes covered by this study, given the rapid sedimentation rates and relatively low plant colonisation, the vast majority (if not all) of the OC is likely to be allochthonous. Whether this represents a new carbon store will depend on it's nature and alternative fate in the absence of the project. For example, aged mineral-associated organic carbon derived from reworked soils or sediments would not represent a new carbon store [79, 80]. However, burial of organic carbon of recent biogenic origin (e.g. phytoplankton or plant debris) may represent a new carbon store if it results in higher preservation rates in comparison to remineralisation in the dynamic estuarine environment. Tools such as biomarkers [81] and stable isotopes are being developed to better identify sources of carbon [19], while size and density fractionation can be used to determine mineral associated organic carbon, and radiocarbon dating for OC age [79, 80]. Combining these with integrated studies of interconnected blue carbon ecosystems across the land-ocean transect would help address the appropriateness of accounting for allochthonous carbon.

## Conclusions

Our results show that at Steart Marshes, fast rates of sediment accumulation and high sediment carbon content combine to result in exceptionally fast carbon accumulation rates. Carbon accumulation at Steart Marshes over the first four years following reinstatement of tidal flow is fifty times larger than the direct carbon costs of site construction. However, there are numerous uncertainties (e.g. origin and alternative fate of carbon, greenhouse gas fluxes) that would need to be resolved in order to move to a fully quantitative carbon budget for restored saltmarshes. Perhaps most importantly, most of this carbon accumulation is likely to be allochthonous, and may not therefore represent newly sequestered carbon. It is particularly urgent to determine the origin and fate of the organic carbon relative to it's fate in the absence of the project.

## Supporting information

**S1 File. It contains S1 & S2 Tables and S1-S4 Figs.**
(PDF)

## Acknowledgments

We thank the Wildfowl and Wetlands Trust, particularly Alys Laver and Tim McGrath, for access to the site and their ongoing enthusiasm and support. We thank Grace Biddle and David McKendry for their work in the laboratory. This study uses data from UK National Tide Gauge Network, owned and operated by the Environment Agency, and provided by the British Oceanographic Data Centre.

## Author Contributions

**Conceptualization:** Hannah L. Mossman, Nigel Pontee, Robert B. Sparkes, Rachel M. Dunk.

**Formal analysis:** Hannah L. Mossman, Katie Born, James Scott, Beatriz Serato, Martin J. P. Sullivan, Rachel M. Dunk.

**Investigation:** Hannah L. Mossman, Colin Hill, Peter J. Lawrence, Stuart Rae, Robert B. Sparkes, Rachel M. Dunk.

**Methodology:** Colin Hill.

**Writing – original draft:** Hannah L. Mossman, Nigel Pontee, Martin J. P. Sullivan, Rachel M. Dunk.

**Writing – review & editing:** Hannah L. Mossman, Nigel Pontee, Katie Born, Colin Hill, Peter J. Lawrence, Stuart Rae, James Scott, Beatriz Serato, Robert B. Sparkes, Martin J. P. Sullivan, Rachel M. Dunk.

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
