## [Decision Letter · Decision Letter 0]

19 Nov 2021

PONE-D-21-32403Rapid carbon accumulation at a saltmarsh restored by managed realignment far exceeds carbon emitted in site constructionPLOS ONE

Dear Dr. Mossman,

Thank you for submitting your manuscript to PLOS ONE. After careful consideration, we feel that it has merit but does not fully meet PLOS ONE’s publication criteria as it currently stands. Therefore, we invite you to submit a revised version of the manuscript that addresses the points raised during the review process.

We look forward to receiving your revised manuscript.

Kind regards,

Daehyun Kim, Ph.D.

Academic Editor

PLOS ONE

Journal Requirements:

3. We note that you have referenced (ie. Bewick et al. [5]) which has currently not yet been accepted for publication. Please remove this from your References and amend this to state in the body of your manuscript: (ie “Bewick et al. [Unpublished]”) as detailed online in our guide for authors

http://journals.plos.org/plosone/s/submission-guidelines#loc-reference-style.

4. We note that Figures 1 and 2 in your submission contain copyrighted images. All PLOS content is published under the Creative Commons Attribution License (CC BY 4.0), which means that the manuscript, images, and Supporting Information files will be freely available online, and any third party is permitted to access, download, copy, distribute, and use these materials in any way, even commercially, with proper attribution. For more information, see our copyright guidelines: http://journals.plos.org/plosone/s/licenses-and-copyright.

a) You may seek permission from the original copyright holder of Figures 1 and 2 to publish the content specifically under the CC BY 4.0 license. 

Additional Editor Comments:

Dear Authors,

I am now ready to provide you with my decision for the submission PONE-D-21-32403, titled "Rapid carbon accumulation at a saltmarsh restored by managed realignment far exceeds carbon emitted in site construction". The two reviewers I invited requested major revisions for the manuscript and I concur with them. Both referees agreed that the research itself is timely and worthwhile, but they still had a number of critical concerns that I would like you to elaborate in your revision. Among the concerns, I especially note the following items:

- more spatially explicit analysis of the data

- data comparability issue

- title and abstract are misleading

- quality of your elevation-change assessment

These are what I found important, but I urge you to improve your original manuscript in other aspects as well. I look forward to receiving a revised version of this work in the near future. Thank you.

Sincerely,

Daehyun Kim, Ph.D.

Academic Editor, Ecology section

Reviewers' comments:

Reviewer's Responses to Questions

**Comments to the Author**

1. Is the manuscript technically sound, and do the data support the conclusions?

Reviewer #1: Partly

Reviewer #2: No

2. Has the statistical analysis been performed appropriately and rigorously? 

Reviewer #1: Yes

Reviewer #2: Yes

3. Have the authors made all data underlying the findings in their manuscript fully available?

Reviewer #1: Yes

Reviewer #2: Yes

4. Is the manuscript presented in an intelligible fashion and written in standard English?

Reviewer #1: Yes

Reviewer #2: Yes

5. Review Comments to the Author

Reviewer #1: This manuscript provides data on the topical activity to breach levee banks to achieve carbon sequestration through re-establishing tidal wetlands. Particular consideration is given to how carbon sequestered compares to the carbon emissions arising from the engineering and monitoring work for the managed realignment, as such quantifications of greenhouse gas emissions are needed for carbon market mechanisms. The manuscript is a worthy and timely contribution which will add to the growing body of knowledge on Blue Carbon. The study includes data from a site with several years of tidal inundation, and a comprehensive data set from airborne bathymetry to ground surveys of soil carbon. The manuscript is well written and presented, and relevant literature considered. I recommend publication, after some revision addressing some concerns and missed opportunities on data analyses as explained below.

The field sampling includes several sites within the study area (Line 126), separated further by elevation differences. Yet in most of the analyses, these sites are combined and an opportunity missed to present differences arising in the development of soil organic carbon stocks subject to the prior land use as well as elevation, which would affect colonising saltmarsh, inundation and sedimentation. This seems an opportunity lost, and I like to encourage the authors to present more spatially explicit findings of their data.

Due to the developments at the site following tidal inundation, soil sampling methods varied (Lines 141 onward), and it should be explained or discussed how this affected the assessment.

For the soil carbon measurements and calculations, international methods (e.g. the Blue Carbon manual) should be applied, which provide established guidance for methods. The calculations for site level carbon (Line 240) appear a bit unusual compared to methods in other Blue Carbon projects.

Clarification is needed on the different approaches used for total carbon (TC) and total organic carbon (TOC). TC data are calculated for all depths sections to 50 cm (Line 157), whereas TOC data are calculated for the depths of the sediment horizon which has accreted over time since tidal inundation (Line 171). To set the soil carbon developments in context with managed realignment, taking the sediment be the horizon of the previous land use is fine (Line 229), but it has to be done for both TC and TOC, otherwise the data are not comparable.

In the course of colonisation and land use change following the opening of the levee bank, a dynamic change of sedimentation/erosion and recolonisation would have occurred and resulted in a differentiated pattern of carbon sequestration. The authors are not exploring this at all, but instead lump all data together across locations, times and depths (Line 227) and also across years (Line 235). The reasoning for considering this amalgamation of data since tidal inundation needs to be added, why was it justified (Line234)? Table 1 includes data from the single sampling times since the breach occurred, and more can be done with the data on increase in TOC over time, and resulting changes in the TOC/TC ratio in the course of recolonisation with saltmarsh.

Figure 5 includes a line which needs some more explanation. Line 401 states it represents an estimate of the construction carbon costs, without an explanation given how the cost is calculation and fitted in with the x- and y-axes of the graph. The legend to the figure states the line shows the carbon accumulation rate needed to break even with cost, which is something different and also not clear how it was derived.

Table 2 is rather wordy and would be more useful if more succinct, so that it can be more useful as guidance for other management realignment projects.

Minor edits:

Line 30: reword ‘…higher although more similar…’

Line 39/40: the first sentence of the Introduction is not very clear

Line 105 ODN = ?

Lines451 and 453 spelling: allochthonous

Reviewer #2: This is certainly an interesting and important study assessing the potential for C credits through a managed realignment project focused on blue C. I do have the following main concerns:

1.) Considering the large uncertainties concerning the role of allochthonous OC and IC contributions, title and abstract of the study are misleading. The discussion on allochthonous and inorganic C should be elaborated on, and resulting caveats/implications should already be mentioned in the Abstract. Compare comments in the pdf file attached.

2.) I don't see if and how organic C loss from the excavated material is being considered /factored into your calculations. Compare comments attached.

3.) More emphasis/discussion on the role of non-CO2 greenhouse-gases, particularly methane, should be included. Compare suggested citations added to the pdf attached.

4.) Methods: I advice to clearly demonstrate the quality of your elevation-change assessment based on Lidar data in the main text (instead of just referring to an Supplementary Figure). This data is crucial.

I am concerned about the inorganic C quantification. Please see comments attached.

5.) Figures/Result presentation: You are comparing C percentages. However, from a C crediting perspective it is more useful to compare C stocks = C densities (C mass / volume). You can have higher C percentage in one soil, but still a smaller or equal C stock (= density). Thus, I advice to change reporting and figures to C stocks or densities.

6. PLOS authors have the option to publish the peer review history of their article (what does this mean?). If published, this will include your full peer review and any attached files.

Reviewer #1: No

Reviewer #2: No

---

## [Author Response · Author response to Decision Letter 0]

1 Jun 2022

Please see attached Response to Reviewers file

---

## [Decision Letter · Decision Letter 1]

5 Aug 2022

PONE-D-21-32403R1Rapid carbon accumulation at a saltmarsh restored by managed realignment exceeded carbon emitted in direct site constructionPLOS ONE

Dear Dr. Mossman,

Thank you for submitting your manuscript to PLOS ONE. After careful consideration, we feel that it has merit but does not fully meet PLOS ONE’s publication criteria as it currently stands. Therefore, we invite you to submit a revised version of the manuscript that addresses the points raised during the review process.

We look forward to receiving your revised manuscript.

Kind regards,

Daehyun Kim, Ph.D.

Academic Editor

PLOS ONE

Journal Requirements:

Additional Editor Comments (if provided):

Dear Authors,

I am now ready to provide you with my editorial recommendation for the revised manuscript (PONE-D-21-32403R1), titled "Rapid carbon accumulation at a saltmarsh restored by managed realignment exceeded carbon emitted in direct site construction". The two reviewers I invited were highly supportive for the value and rigor of this research, and I concur with them. Hence, I believe that this manuscript will be publishable after a set minor revisions as suggested by the referees. You will find below a number of constructive comments and questions and I would like you to address each of them in depth.

Sincerely,

Daehyun, Ph.D.

Academic Editor of Ecology section

Reviewers' comments:

Reviewer's Responses to Questions

**Comments to the Author**

1. If the authors have adequately addressed your comments raised in a previous round of review and you feel that this manuscript is now acceptable for publication, you may indicate that here to bypass the “Comments to the Author” section, enter your conflict of interest statement in the “Confidential to Editor” section, and submit your "Accept" recommendation.

Reviewer #3: All comments have been addressed

Reviewer #4: (No Response)

2. Is the manuscript technically sound, and do the data support the conclusions?

Reviewer #3: Partly

Reviewer #4: Yes

3. Has the statistical analysis been performed appropriately and rigorously? 

Reviewer #3: Yes

Reviewer #4: Yes

4. Have the authors made all data underlying the findings in their manuscript fully available?

Reviewer #3: Yes

Reviewer #4: (No Response)

5. Is the manuscript presented in an intelligible fashion and written in standard English?

Reviewer #3: Yes

Reviewer #4: Yes

6. Review Comments to the Author

Reviewer #3: Review of PONE-D-21-32403R1 “ Rapid carbon accumulation at a saltmarsh restored by managed realignment exceeded carbon emitted in direct site construction”

This paper presents a well-written analysis of sediment carbon dynamics associated with a managed realignment site. Of particular interest, the authors estimate the carbon emission value of site construction activities, a component of carbon budgeting that is rarely addressed in such studies. The combination of on the ground data collection and remote sensing-based approaches provides for a high spatial and temporal resolution analysis of change over time in sediment surface elevation. These data, in combination with carbon content analysis of sediment cores collected within the restored area are used to estimate the total carbon stock of the newly deposited sediments within the restored area. The authors do an admirable job of describing and justifying their methodology.

The one criticism I have of this manuscript is that the interpretation leaves the reader wondering about how important the carbon burial in this system is. The discussion seems to oscillate between suggesting that the site could have a huge carbon benefit and that there’s really no way to tell how important this site is as a sink for carbon without an extensive amount of additional research – leading the reader to come to their own conclusions.

The authors note minimal revegetation of the site for much of the collection period and major importing of sediments. The evidence seems to indicate that the majority of carbon accumulated at this site is allochthonous which would be consistent with other managed realignment studies cited in this manuscript. I’d suggest that the allochthonous fraction will likely offset MOST of the carbon accumulation benefits of the site.

To claim that there is a benefit to importing and burying allochthonous carbon it is necessary to demonstrate that the carbon would have otherwise been oxidized. Unless that can be proven conclusively, the carbon burial at this site merely represents a lateral transfer of C from other nearby ecosystems, not a net benefit (and therefore not an offset to carbon emitted due to site construction). Certainly, over time, this site will produce net emission offsets as the vegetation grows in and begins burying carbon in place (assuming as noted, that methane production isn’t significant here). The sediment carbon profiles suggest that this is not yet occurring at this site.

Ultimately, I think the data presented here, combined with similar results from other similar sites (the Bay of Fundy study reference here and Drexel et al 2020.) demonstrate clearly the significant role of allochthonous carbon at sites that are importing sediments. For accounting and crediting purposes, the most conservative approach would be to consider this the “background” carbon against which future additions can be measured. The difference between the C in sediment of the project area and that of the surrounding salt marsh (multiplied by the total sediment volume of the project area) could provide a reasonable estimate of the likely carbon burial value of this site if one assumes that given enough time, it will reach equivalence with the natural marsh.

There is valuable data in the manuscript which will be of interest to PLOS readers but I feel like this contribution could be more impactful by more strongly emphasizing the allochthonous nature of the measured carbon and how what that means for its carbon value.

Note a few minor grammatical errors/misspellings:

82 “ including changes gas fluxes following”

330 “eroision”

331 “expecially”

Reviewer #4: The authors provide what I consider to be a novel approach to look at C accumulation rates from a specific type of restoration – managed realignment, and compare these to the CO2 emitted from the actual effort required to restore the site. This type of information is needed to guide future restoration techniques to determine if the restoration is successful at returning C and to determine the environmental cost of that restoration. I think their approach is sound. At first I wasn’t sure why you didn’t measure the change in the silt layer above the previous land use in thee cores to calculate C accumulation, but then realized that the spatial representation is much better from the elevation models. The only concern that I have is I would have preferred to have seen an intact site nearby if possible to act as a reference. It sounded to me like the natural reference site they used was one that was also a developing site, it’s described as a pioneer salt marsh that was bare with some Spartina anglica present. Using an intact reference site would have allowed them to determine if the initial pulse of sediment (0.449 m yr-1) in Table 1 was due to the low elevation of the restored site or an unusually high sediment load that occurred that year. Can you address this in the text as a caveat? Also, would it be possible to include the sedimentation rates from this natural site in Table 1?

Double check some of the numbers. The mileage for the restoration was 69,000 km? That’s a lot. Also, based on the numbers you present, I estimated that the amount of carbon returned was only 50x greater than the amount of C used to restore the site, not 2 orders. But I could have missed something there.

Other than that, I only had a few suggestions below in the introduction. Overall, I thought it was a well written paper and I enjoyed reviewing it.

Rich MacKenzie

L43 Not all ecosystems contain substantial carbon stocks (see Alongi papers). I would suggest changing this to “can contain substantial carbon stocks”.

L43 the way this is currently written, it reads as if carbon stocks currently store carbon out of the atmosphere. Plus, you say this already in the first sentence. I suggest changing to: They can also contain substantial carbon stocks largely derived from atmospheric carbon and these stocks are sensitive to changes in climate or land use.

L46 sequestered is associated with fixing CO2 from the atmosphere. Consider “Both allochthonous and autochthonous carbon accumulate in saltmarshes, through the deposition of sediment and organic matter carried in by tides or through plant growth, respectively. Or something like that.

L49 you haven’t really gone into their importance aside from C. Consider Despite their large carbon stocks, ~50%...

L51 degraded from what?

L57-59 Carbon as been recognized as a benefit of saltmarsh restoration. Wouldn’t this provide a further motivation for restoration instead of creation? Changing creation to restoration also ties this paragraph into the next one.

L60 accumulation in restored

L60 delete Robustly. I think just any attempts to quantify the rate of carbon accumulation is badly needed!

L64-68 This is great and I completely agree. I also think comparing different restoration techniques at restoring C is needed. This is a big issue with mangroves. I would consider adding something like that here as well, that comparing different methods is needed, which is sort of the goal of this paper, to assess managed realignment.

L69-75 using radionuclides (210Pb) to date sediments is another method that is popularly used to quantify C accumulation (Drexler et al. 2019; Arias-Ortez et al. 2018; etc.)

L82 changes in gas fluxes

L81-87 – yes!! Great point that a lot of restoration projects fail to consider.

L142 what type of soil auger?

L143 I am not familiar with the term surface silts. COnsier changing to surface sediments?

L149-150 how did you sample the sediment plate? I am also not familiar with the term sediment plate

L256 can you please clarify that the bulk density was specific to the silt layer that resulted after the restoration?

L380 wow! 69,573 km? Is that correct? That’s almost 7 round trips between London and New York City.

L392 – on L373 you state that 34,672 tC has accumulated since the project, and in L388 that the estimated total construction emissions were 753 tC. Unless I missed something, two orders would 75,300. The amount of carbon that has accumulated is 50x.

L395-400 can you explain possible mechanisms as to why you are seeing these differences? Are total suspended sediment greater near Steart Marshes than those other sites?

L496-502 – what were the salinities at your site? This would support or not support this statement that restoration can increase CH4 if salinities are high enough.

7. PLOS authors have the option to publish the peer review history of their article (what does this mean?). If published, this will include your full peer review and any attached files.

Reviewer #3: No

Reviewer #4: **Yes: **Richard A. MacKenzie

---

## [Author Response · Author response to Decision Letter 1]

19 Sep 2022

Please see the response to reveiwers document

---

## [Editor Report · Decision Letter 2]

23 Sep 2022

Rapid carbon accumulation at a saltmarsh restored by managed realignment exceeded carbon emitted in direct site construction

PONE-D-21-32403R2

Dear Dr. Mossman,

We’re pleased to inform you that your manuscript has been judged scientifically suitable for publication and will be formally accepted for publication once it meets all outstanding technical requirements.

Kind regards,

Daehyun Kim, Ph.D.

Academic Editor

PLOS ONE

Additional Editor Comments (optional):

I evaluate that the authors did an admirably good job in addressing all questions and concerns raised by the two referees I invited for the second-round review. I feel no need to send out this revised version for another round of external review. Therefore, I am happy to accept this fine work for publication in PLOS ONE. I believe that this research will be of wide and deep interest to many scholars studying global climate change and carbon sequestration in coastal ecosystems. I look forward to seeing more papers of this sort by the authors.
---

## [Editor Report · Acceptance letter]

8 Nov 2022

PONE-D-21-32403R2 

Rapid carbon accumulation at a saltmarsh restored by managed realignment exceeded carbon emitted in direct site construction 

Dear Dr. Mossman:

I'm pleased to inform you that your manuscript has been deemed suitable for publication in PLOS ONE. Congratulations! Your manuscript is now with our production department. 

Kind regards, 

on behalf of

Dr. Daehyun Kim 

Academic Editor

PLOS ONE